# LEARNING HOW TO EXPLAIN NEURAL NETWORKS: PATTERNNET AND PATTERNATTRIBUTION

**Pieter-Jan Kindermans**[*]
Google Brain
pikinder@google.com

**Kristof T. Schütt & Maximilian Alber**
TU Berlin
{kristof.schuett,maximilian.alber}@tu-berlin.de

**Klaus-Robert Müller**[†]
TU Berlin
klaus-robert.mueller@tu-berlin.de

**Dumitru Erhan & Been Kim**
Google Brain
{dumitru,beenkim}@google.com

**Sven Dähne**[‡]
TU Berlin
sven.daehne@tu-berlin.de

## ABSTRACT

DeConvNet, Guided BackProp, LRP, were invented to better understand deep neural networks. We show that these methods do not produce the theoretically correct explanation for a linear model. Yet they are used on multi-layer networks with millions of parameters. This is a cause for concern since linear models are simple neural networks. We argue that explanation methods for neural nets should work reliably in the limit of simplicity, the linear models. Based on our analysis of linear models we propose a generalization that yields two explanation techniques (PatternNet and PatternAttribution) that are theoretically sound for linear models and produce improved explanations for deep networks.

## 1 INTRODUCTION

Deep learning made a huge impact on a wide variety of applications (Krizhevsky et al., 2012; Sutskever et al., 2014; LeCun et al., 2015; Schmidhuber, 2015; Mnih et al., 2015; Silver et al., 2016) and recent neural network classifiers have become excellent at detecting relevant *signals* (e.g., the presence of a cat) contained in input data points such as images by filtering out all other, non-relevant and *distracting* components also present in the data. This separation of signal and distractors is achieved by passing the input through many layers with millions of parameters and nonlinear activation functions in between, until finally at the output layer, these models yield a highly condensed version of the signal, e.g. a single number indicating the probability of a cat being in the image.

While deep neural networks learn efficient and powerful representations, they are often considered a 'black-box'. In order to better understand classifier decisions and to gain insight into how these models operate, a variety techniques have been proposed (Simonyan et al., 2014; Yosinski et al., 2015; Nguyen et al., 2016; Baehrens et al., 2010; Bach et al., 2015; Montavon et al., 2017; Zeiler & Fergus, 2014; Springenberg et al., 2015; Zintgraf et al., 2017; Sundararajan et al., 2017; Smilkov et al., 2017). These methods for explaining classifier decisions operate under the assumption that it is possible to propagate the condensed output signal back through the classifier to arrive at something that shows how the relevant signal was encoded in the input and thereby explains the classifier decision. Simply put, if the classifier detected a cat, the visualization should point to the cat-relevant aspects of the input image from the perspective of the network. Techniques that are based on this principle include saliency maps from network gradients (Baehrens et al., 2010; Simonyan et al., 2014), DeConvNet (Zeiler & Fergus, 2014, DCN), Guided BackProp (Springenberg et al., 2015, GBP),

---

[*]Part of this work was done at TU Berlin, part of the work was part of the Google Brain Residency program.
[†]KRM is also with Korea University and Max Planck Institute for Informatics, Saarbrücken, Germany
[‡]Sven Dähne is now at Amazon

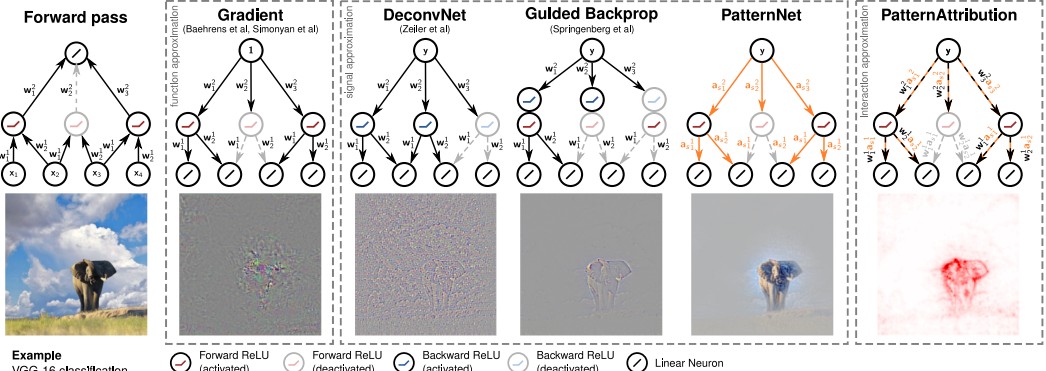

Figure 1: Illustration of explanation approaches. Function and signal approximators visualize the explanation using the original color channels. The attribution is visualized as a heat map of pixel-wise contributions to the output

Layer-wise Relevance Propagation (Bach et al., 2015, LRP) and the Deep Taylor Decomposition (Montavon et al., 2017, DTD), Integrated Gradients (Sundararajan et al., 2017) and SmoothGrad (Smilkov et al., 2017).

The merit of explanation methods is often demonstrated by applying them to state-of-the-art deep learning models in the context of high dimensional real world data, such as ImageNet, where the provided explanation is intuitive to humans. Unfortunately, theoretical analysis as well as quantitative empirical evaluations of these methods are lacking.

Deep neural networks are essentially a composition of linear transformations connected with non-linear activation functions. Since approaches, such as DeConvNet, Guided BackProp, and LRP, back-propagate the explanations in a layer-wise fashion, it is crucial that the individual linear layers are handled correctly. In this work we show that these gradient-based methods *fail* to recover the signal even for a single-layer architecture, i.e. a linear model. We argue that therefore they cannot be expected to reliably explain a deep neural network and demonstrate this with quantitative and qualitative experiments. In particular, we provide the following key contributions:

- We analyze the performance of existing explanation approaches in the controlled setting of a linear model (Sections 2 and 3).

- We categorize explanation methods into three groups – functions, signals and attribution (see Fig. 1) – that require fundamentally different interpretations and are complementary in terms of information about the neural network (Section 3).

- We propose two novel explanation methods – PatternNet and PatternAttribution – that alleviate shortcomings of current approaches, as discovered during our analysis, and improve explanations in real-world deep neural networks visually and quantitatively (Sections 4 and 5).

This presents a step towards a thorough analysis of explanation methods and suggests qualitatively and measurably improved explanations. These are crucial requirements for reliable explanation techniques, in particular in domains, where explanations are not necessarily intuitive, e.g. in health and the sciences Schütt et al. (2017).

**Notation and scope** Scalars are lowercase letters ($i$), column vectors are bold ($\boldsymbol{u}$), element-wise multiplication is ($\odot$). The covariance between $\boldsymbol{u}$ and $\boldsymbol{v}$ is $\text{cov}[\boldsymbol{u}, \boldsymbol{v}]$, the covariance of $\boldsymbol{u}$ and $i$ is $\text{cov}[\boldsymbol{u}, i]$. The variance of a scalar random variable $i$ is $\sigma_i^2$. Estimates of random variables will have a hat ($\hat{\boldsymbol{u}}$). We analyze neural networks excluding the final soft-max output layer. To allow for analytical treatment, we only consider networks with linear neurons optionally followed by a rectified linear unit (ReLU), max-pooling or soft-max. We analyze linear neurons and nonlinearities independently such that every neuron has its own weight vector. These restrictions are similar to those in the saliency map (Simonyan et al., 2014), DCN (Zeiler & Fergus, 2014), GBP (Springenberg

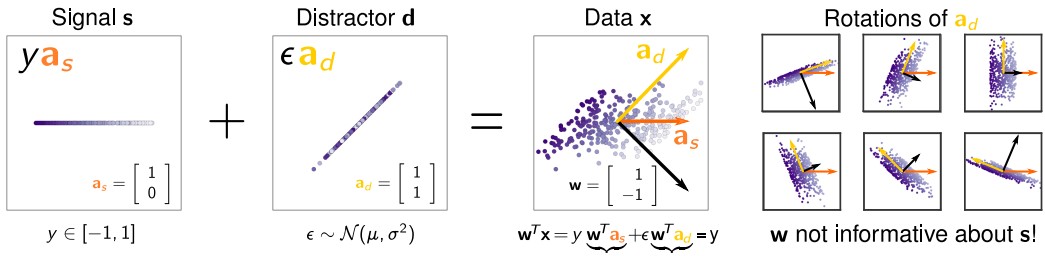

Figure 2: For linear models, i.e., a simple neural network, the weight vector does **not** explain the signal it detects Haufe et al. (2014). The data $x = ya_s + \epsilon a_d$ is color-coded w.r.t. the output $y = w^T x$. Only the signal $s = ya_s$ contributes to $y$. The weight vector $w$ does not agree with the signal direction, since its primary objective is canceling the distractor. Therefore, rotations of the basis vector $a_d$ of the distractor with constant signal $s$ lead to rotations of the weight vector (**right**).

et al., 2015), LRP (Bach et al., 2015) and DTD (Montavon et al., 2017). Without loss of generality, biases are considered constant neurons to enhance clarity.

## 2 Understanding linear models

In this section, we analyze explanation methods for deep neural network, starting with the simplest neural network setting: a purely linear model and data sampled from a linear generative model. This setup allows us to (i) fully control how signal and distractor components are encoded in the input data and (ii) analytically track how the resulting explanation relates to the known signal component. This analysis allows us then to highlight shortcomings of current explanation approaches that carry over to deep neural networks.

Consider the following toy example (see Fig. 2) where we generate data $x$ as:

$$x = s + d \qquad s = a_s y, \qquad \text{with } a_s = (1, 0)^T, \qquad y \in [-1, 1]$$
$$d = a_d \epsilon, \qquad \text{with } a_d = (1, 1)^T, \qquad \epsilon \sim \mathcal{N}\left(\mu, \sigma^2\right).$$

We train a linear regression model to extract $y$ from $x$. By construction, $s$ is the *signal* in our data, i.e., the part of $x$ containing information about $y$. Using the terminology of Haufe et al. (2014) the *distractor* $d$ obfuscates the signal making the detection task more difficult. To optimally extract $y$, our model has to be able to filter out the distractor $d$. This is why the weight vector is also called the *filter*. In the example, $w = [1, -1]^T$ fulfills this convex task.

From this example, we can make several observations: The optimal weight vector $w$ does *not* align, in general, with the signal direction $a_s$, but tries to filter the contribution of the distractor (see Fig. 2). This is optimally solved when the weight vector is orthogonal to the distractor $w^T d = 0$. Therefore, when the direction of the distractor $a_d$ changes, $w$ must follow, as illustrated on the right hand side of the figure. On the other hand, a change in signal direction $a_s$ can be compensated for by a change in sign and magnitude of $w$ such that $w^T a_s = 1$, but the direction stays constant.

The fact that the direction of the weight vector in a linear model is largely determined by the distractor implies that given only the weight vector, we cannot know what part of the input produces the output $y$. On the contrary, the direction $a_s$ must be learned from data.

Now assume that we have additive isotropic Gaussian noise. The mean of the noise can easily be compensated for with a bias change. Therefore, we only have to consider the zero-mean case. Since isotropic Gaussian noise does not contain any correlations or structure, the only way to remove it is by averaging over different measurements. It is *not* possible to cancel it out effectively by using a well-chosen weight vector. However, it is well known that adding Gaussian noise shrinks the weight vector and corresponds to L2 regularization. In the absence of a structured distractor, the smallest weight vector $w$ such that $w^T a_s = 1$ is the one in the direction of the signal. Therefore in practice both these effects influence the actual weight vector.

As already indicated above, deep neural networks are essentially a composition of linear layers and non-linear activation functions. In the next section, we will show that gradient-based methods, e.g., DeConvNet, Guided BackProp, and LRP, are not able to distinguish signal from distractor in a linear model and therefore back-propagate sub-optimal explanations in deeper networks. This analysis allows us to develop improved layer-wise explanation techniques and to demonstrate quantitative and qualitative better explanations for deep neural networks.

**Terminology**   Throughout this manuscript we will use the following terminology: The *filter* $\boldsymbol{w}$ tells us how to extract the output $y$ optimally from data $\boldsymbol{x}$. The *pattern* $\boldsymbol{a}_s$ is the direction in the data along which the desired output $y$ varies. Both constitute the *signal* $\boldsymbol{s} = \boldsymbol{a}_s y$, i.e., the contributing part of $\boldsymbol{x}$. The *distractor* $\boldsymbol{d}$ is the component of the data that does not contain information about the desired output.

## 3   OVERVIEW OF EXPLANATION APPROACHES AND THEIR BEHAVIOR

In this section, we take a look at a subset of explanation methods for individual classifier decisions and discuss how they are connected to our analysis of linear models in the previous section. Fig. 1 gives an overview of the different types of explanation methods which can be divided into function, signal and attribution visualizations. These three groups all present different information about the network and complement each other.

**Functions – gradients, saliency map**   Explaining the function in input space corresponds to describing the operations the model uses to extract $y$ from $\boldsymbol{x}$. Since deep neural networks are highly nonlinear, this can only be approximated. The saliency map estimates how moving along a particular direction in input space influences $y$ (i.e., sensitivity analysis) where the direction is given by the model gradient (Baehrens et al., 2010; Simonyan et al., 2014). In case of a linear model $y = \boldsymbol{w}^T \boldsymbol{x}$, the saliency map reduces to analyzing the weights $\partial y / \partial \boldsymbol{x} = \boldsymbol{w}$. Since it is mostly determined by the distractor, as demonstrated above, it is not representing the signal. It tells us how to extract the signal, not what the signal is in a deep neural network.

**Signal – DeConvNet, Guided BackProp, PatternNet**   The signal $\boldsymbol{s}$ detected by the neural network is the component of the data that caused the networks activations. Zeiler & Fergus (2014) formulated the goal of these methods as "[...] to map these activities back to the input pixel space, showing what input pattern originally caused a given activation in the feature maps".

In a linear model, the signal corresponds to $\boldsymbol{s} = \boldsymbol{a}_s y$. The pattern $\boldsymbol{a}_s$ contains the signal direction, i.e., it tells us where a change of the output variable is expected to be measurable in the input (Haufe et al., 2014). Attempts to visualize the signal for deep neural networks were made using DeConvNet (Zeiler & Fergus, 2014) and Guided BackProp (Springenberg et al., 2015). These use the same algorithm as the saliency map, but treat the rectifiers differently (see Fig. 1): DeConvNet leaves out the rectifiers from the forward pass, but adds additional ReLUs after each deconvolution, while Guided BackProp uses the ReLUs from the forward pass as well as additional ones. The back-projections for the linear components of the network correspond to a superposition of what are assumed to be the signal directions of each neuron. For this reason, these projections must be seen as an approximation of the features that activated the higher layer neuron. It is not a reconstruction in input space (Zeiler & Fergus, 2014).

For the simplest of neural networks – the linear model – these visualizations reduce to the gradient[1]. They show the filter $\boldsymbol{w}$ and *neither* the pattern $\boldsymbol{a}_s$, *nor* the signal $\boldsymbol{s}$. Hence, DeConvNet and Guided BackProp do not guarantee to produce the detected signal for a linear model, which is proven by our toy example in Fig. 2. Since they do produce compelling visualizations, we will later investigate whether the direction of the filter $\boldsymbol{w}$ coincides with the direction of the signal $\boldsymbol{s}$. We will show that this is *not* the case and propose a new approach, PatternNet (see Fig. 1), to estimate the correct direction that improves upon the DeConvNet and Guided BackProp visualizations.

---

[1]In tensorflow terminoloy: linear model on MNIST can be seen as a convolutional neural network with VALID padding and a 28 by 28 filter size.

**Attribution – LRP, Deep Taylor Decomposition, PatternAttribution**   Finally, we can look at how much the signal dimensions contribute to the output through the layers. This will be referred to as the *attribution*. For a linear model, the optimal attribution would be obtained by element-wise multiplying the signal with the weight vector: $\boldsymbol{r}^{input} = \boldsymbol{w} \odot \boldsymbol{a}y$, with $\odot$ the element-wise multiplication. Bach et al. (2015) introduced *layer-wise relevance propagation* (LRP) as a decomposition of pixel-wise contributions (called *relevances*). Montavon et al. (2017) extended this idea and proposed the deep Taylor decomposition (DTD). The key idea of DTD is to decompose the activation of a neuron in terms of contributions from its inputs. This is achieved using a first-order Taylor expansion around a root point $\boldsymbol{x}_0$ with $\boldsymbol{w}^T \boldsymbol{x}_0 = 0$. The relevance of the selected output neuron $i$ is initialized with its output from the forward pass. The relevance from neuron $i$ in layer $l$ is re-distributed towards its input as:

$$r_i^{output} = y, \qquad r_{j \neq i}^{output} = 0, \qquad \boldsymbol{r}^{l-1,i} = \frac{\boldsymbol{w} \odot (\boldsymbol{x} - \boldsymbol{x}_0)}{\boldsymbol{w}^T \boldsymbol{x}} r_i^l.$$

To obtain the relevance for neuron $i$ in layer $l-1$ the incoming relevances from all connected neurons $j$ in layer $l$ are summed

$$r_i^{l-1} = \sum_j r_i^{l-1,j}.$$

Here we can safely assume that $\boldsymbol{w}^T \boldsymbol{x} > 0$ because a non-active ReLU unit from the forward pass stops the re-distribution in the backward pass. This is identical to how a ReLU stops the propagation of the gradient. The difficulty in the application of the deep Taylor decomposition is the choice of the root point $\boldsymbol{x}_0$, for which many options are available. It is important to recognize at this point that selecting a root point for the DTD corresponds to estimating the distractor $\boldsymbol{x}_0 = \boldsymbol{d}$ and, by that, the signal $\hat{\boldsymbol{s}} = \boldsymbol{x} - \boldsymbol{x}_0$. PatternAttribution is a DTD extension that learns from data how to set the root point.

Summarizing, the **function** extracts the **signal** from the data by removing the distractor. The **attribution** of output values to input dimensions shows how much an individual component of the signal contributes to the output, which is what LRP calls *relevance*.

## 4   LEARNING TO ESTIMATE THE SIGNAL

Visualizing the function has proven to be straightforward (Baehrens et al., 2010; Simonyan et al., 2014). In contrast, visualizing the signal (Haufe et al., 2014; Zeiler & Fergus, 2014; Springenberg et al., 2015) and the attribution (Bach et al., 2015; Montavon et al., 2017; Sundararajan et al., 2017) is more difficult. It requires a good estimate of what is the signal and what is the distractor. In the following section we first propose a quality measure for neuron-wise signal estimators. This allows us to evaluate existing approaches and, finally, derive signal estimators that optimize this criterion. These estimators will then be used to explain the signal (PatternNet) and the attribution (PatternAttribution). All mentioned techniques as well as our proposed signal estimators treat neurons independently, i.e., the full explanation will be a superposition of neuron-wise explanations.

### 4.1   QUALITY CRITERION FOR SIGNAL ESTIMATORS

Recall that the input data $\boldsymbol{x}$ comprises both signal and distractor: $\boldsymbol{x} = \boldsymbol{s} + \boldsymbol{d}$, and that the signal contributes to the output but the distractor does not. Assuming the filter $\boldsymbol{w}$ has been trained sufficiently well to extract $y$, we have

$$\boldsymbol{w}^T \boldsymbol{x} = y, \quad \boldsymbol{w}^T \boldsymbol{s} = y, \quad \boldsymbol{w}^T \boldsymbol{d} = 0.$$

Note that estimating the signal based on these conditions alone is an ill-posed problem. We could limit ourselves to linear estimators of the form $\hat{\boldsymbol{s}} = \boldsymbol{u}(\boldsymbol{w}^T \boldsymbol{u})^{-1} y$, with $\boldsymbol{u}$ a random vector such that $\boldsymbol{w}^T \boldsymbol{u} \neq 0$. For such an estimator, the signal estimate $\hat{\boldsymbol{s}} = \boldsymbol{u} \left( \boldsymbol{w}^T \boldsymbol{u} \right)^{-1} y$ satisfies $\boldsymbol{w}^T \hat{\boldsymbol{s}} = y$. This implies the existence of an infinite number of possible rules for the DTD as well as infinitely many back-projections for the DeConvNet family.

To alleviate this issue, we introduce the following quality measure $\rho$ for a signal estimator $S(\boldsymbol{x}) = \hat{\boldsymbol{s}}$ that will be written with explicit variances and covariances using the shorthands $\hat{\boldsymbol{d}} = \boldsymbol{x} - S(\boldsymbol{x})$ and

$y = \boldsymbol{w}^T \boldsymbol{x}$:

$$\rho(S) = 1 - \max_{\boldsymbol{v}} corr\left(\boldsymbol{w}^T \boldsymbol{x}, \boldsymbol{v}^T\left(\boldsymbol{x} - S(\boldsymbol{x})\right)\right) = 1 - \max_{\boldsymbol{v}} \frac{\boldsymbol{v}^T \text{cov}[\hat{\boldsymbol{d}}, y]}{\sqrt{\sigma_{\boldsymbol{v}^T \hat{\boldsymbol{d}}}^2 \sigma_y^2}}. \tag{1}$$

This criterion introduces an additional constraint by measuring how much information about $y$ can be reconstructed from the residuals $\boldsymbol{x} - \hat{\boldsymbol{s}}$ using a linear projection. The best signal estimators remove most of the information in the residuals and thus yield large $\rho(S)$. Since the correlation is invariant to scaling, we constrain $\boldsymbol{v}^T \hat{\boldsymbol{d}}$ to have variance $\sigma_{\boldsymbol{v}^T \hat{\boldsymbol{d}}}^2 = \sigma_y^2$. Finding the optimal $\boldsymbol{v}$ for a fixed $S(\boldsymbol{x})$ amounts to a least-squares regression from $\hat{\boldsymbol{d}}$ to $y$. This enables us to assess the quality of signal estimators efficiently.

## 4.2   EXISTING SIGNAL ESTIMATORS

Let us now discuss two signal estimators that have been used in previous approaches.

$S_{\boldsymbol{x}}$ – **the identity estimator**    The naive approach to signal estimation is to assume the entire data is signal and there are no distractors:

$$S_{\boldsymbol{x}}(\boldsymbol{x}) = \boldsymbol{x}.$$

With this being plugged into the deep Taylor framework, we obtain the $z$-rule (Montavon et al., 2017) which is equivalent to LRP (Bach et al., 2015). For a linear model, this corresponds to $\boldsymbol{r} = \boldsymbol{w} \odot \boldsymbol{x}$ as the attribution. It can be shown that for ReLU and max-pooling networks, the $z$-rule reduces to the element-wise multiplication of the input and the saliency map (Shrikumar et al., 2016; Kindermans et al., 2016). This means that for a whole network, the assumed signal is simply the original input image. It also implies that, if there are distractors present in the data, they are included in the attribution:

$$\boldsymbol{r} = \boldsymbol{w} \odot \boldsymbol{x} = \boldsymbol{w} \odot \boldsymbol{s} + \boldsymbol{w} \odot \boldsymbol{d}.$$

When moving through the layers by applying the filters $\boldsymbol{w}$ during the forward pass, the contributions from the distractor $\boldsymbol{d}$ are cancelled out. However, they cannot be cancelled in the backward pass by the element-wise multiplication. The distractor contributions $\boldsymbol{w} \odot \boldsymbol{d}$ that are included in the LRP explanation cause the noisy nature of the visualizations based on the $z$-rule.

$S_{\boldsymbol{w}}$ – **the filter based estimator**    The implicit assumption made by DeConvNet and Guided Back-Prop is that the detected signal varies in the direction of the weight vector $\boldsymbol{w}$. This weight vector has to be normalized in order to be a valid signal estimator. In the deep Taylor decomposition framework this corresponds to the $\boldsymbol{w}^2$-rule and results in the following signal estimator:

$$S_{\boldsymbol{w}}(\boldsymbol{x}) = \frac{\boldsymbol{w}}{\boldsymbol{w}^T \boldsymbol{w}} \boldsymbol{w}^T \boldsymbol{x}.$$

For a linear model, this produces an attribution of the form $\frac{\boldsymbol{w} \odot \boldsymbol{w}}{\boldsymbol{w}^T \boldsymbol{w}} y$. This estimator does not reconstruct the proper signal in the toy example of section 2. Empirically it is also sub-optimal in our experiment in Fig. 3.

## 4.3   PATTERNNET AND PATTERNATTRIBUTION

We suggest to learn the signal estimator $S$ from data by optimizing the previously established criterion. A signal estimator $S$ is optimal with respect to Eq. (1) if the correlation is zero for all possible $\boldsymbol{v}$: $\forall \boldsymbol{v}, \text{cov}[y, \hat{\boldsymbol{d}}]\boldsymbol{v} = \boldsymbol{0}$. This is the case when there is no covariance between $y$ and $\hat{\boldsymbol{d}}$. Because of linearity of the covariance and since $\hat{\boldsymbol{d}} = \boldsymbol{x} - S(\boldsymbol{x})$ the above condition leads to

$$\text{cov}[y, \hat{\boldsymbol{d}}] = \boldsymbol{0} \Rightarrow \text{cov}[\boldsymbol{x}, y] = \text{cov}[S(\boldsymbol{x}), y]. \tag{2}$$

It is important to recognize that the covariance is a summarizing statistic and consequently the problem can still be solved in multiple ways. We will present two possible solutions to this problem. Note that when optimizing the estimator, the contribution from the bias neuron will be considered 0 since it does not covary with the output $y$.

$S_a$ – **The linear estimator**   A linear neuron can only extract linear signals $s$ from its input $x$. Therefore, we could assume a linear dependency between $s$ and $y$, yielding a signal estimator:

$$S_a(x) = a w^T x. \tag{3}$$

Plugging this into Eq. (2) and optimising for $a$ yields

$$\text{cov}[x, y] = \text{cov}[a w^T x, y] = a\text{cov}[y, y] \Rightarrow a = \frac{\text{cov}[x, y]}{\sigma_y^2}. \tag{4}$$

Note that this solution is equivalent to the approach commonly used in neuro-imaging (Haufe et al., 2014) despite different derivation. With this approach we can recover the signal of our toy example in section 2. It is equivalent to the filter-based approach only if the distractors are orthogonal to the signal. We found that the linear estimator works well for the convolutional layers. However, when using this signal estimator with ReLUs in the dense layers, there is still a considerable correlation left in the distractor component (see Fig. 3).

$S_{a_{+-}}$ – **The two-component estimator**   To move beyond the linear signal estimator, it is crucial to understand how the rectifier influences the training. Since the gate of the ReLU closes for negative activations, the weights only need to filter the distractor component of neurons with $y > 0$. Since this allows the neural network to apply filters locally, we cannot assume a global distractor component. We rather need to distinguish between the positive and negative regime:

$$x = \begin{cases} s_+ + d_+ & \text{if } y > 0 \\ s_- + d_- & \text{otherwise} \end{cases}$$

Even though signal and distractor of the negative regime are canceled by the following ReLU, we still need to make this distinction in order to approximate the signal. Otherwise, information about whether a neuron fired would be retained in the distractor. Thus, we propose the two-component signal estimator:

$$S_{a_{+-}}(x) = \begin{cases} a_+ w^T x, & \text{if } w^T x > 0 \\ a_- w^T x, & \text{otherwise} \end{cases} \tag{5}$$

Next, we derive expressions for the patterns $a_+$ and $a_-$. We denote expectations over $x$ within the positive and negative regime with $\mathbb{E}_+[x]$ and $\mathbb{E}_-[x]$, respectively. Let $\pi_+$ be the expected ratio of inputs $x$ with $w^T x > 0$. The covariance of data/signal and output become:

$$\text{cov}[x, y] = \quad \pi_+ \left( \mathbb{E}_+[xy] - \mathbb{E}_+[x] \mathbb{E}[y] \right) + \quad (1 - \pi_+) \left( \mathbb{E}_-[xy] - \mathbb{E}_-[x] \mathbb{E}[y] \right) \tag{6}$$

$$\text{cov}[s, y] = \quad \pi_+ \left( \mathbb{E}_+[sy] - \mathbb{E}_+[s] \mathbb{E}[y] \right) + \quad (1 - \pi_+) \left( \mathbb{E}_-[sy] - \mathbb{E}_-[s] \mathbb{E}[y] \right) \tag{7}$$

Assuming both covariances are equal, we can treat the positive and negative regime separately using Eq. (2) to optimize the signal estimator:

$$\mathbb{E}_+[xy] - \mathbb{E}_+[x] \mathbb{E}[y] \quad = \quad \mathbb{E}_+[sy] - \mathbb{E}_+[s] \mathbb{E}[y]$$

Plugging in Eq. (5) and solving for $a_+$ yields the required parameter ($a_-$ analogous).

$$a_+ \quad = \quad \frac{\mathbb{E}_+[xy] - \mathbb{E}_+[x] \mathbb{E}[y]}{w^T \mathbb{E}_+[xy] - w^T \mathbb{E}_+[x] \mathbb{E}[y]} \tag{8}$$

The solution for $S_{a_{+-}}$ reduces to the linear estimator when the relation between input and output is linear. Therefore, it solves our introductory linear example correctly.

**PatternNet and PatternAttribution**   Based on the presented analysis, we propose PatternNet and PatternAttribution as illustrated in Fig. 1. *PatternNet* yields a layer-wise back-projection of the estimated signal to input space. The signal estimator is approximated as a superposition of neuron-wise, nonlinear signal estimators $S_{a_{+-}}$ in each layer. It is equal to the computation of the gradient where during the backward pass the weights of the network are replaced by the informative directions. In Fig. 1, a visual improvement over DeConvNet and Guided Backprop is apparent.

*PatternAttribution* exposes the attribution $w \odot a_+$ and improves upon the layer-wise relevance propagation (LRP) framework (Bach et al., 2015). It can be seen as a root point estimator for the Deep-Taylor Decomposition (DTD). Here, the explanation consists of neuron-wise contributions of the estimated *signal* to the classification score. By ignoring the distractor, PatternAttribution can reduce the noise and produces much clearer heat maps. By working out the back-projection steps in the Deep-Taylor Decomposition with the proposed root point selection method, it becomes obvious that PatternAttribution is also analogous to the backpropagation operation. In this case, the weights are replaced during the backward pass by $w \odot a_+$.

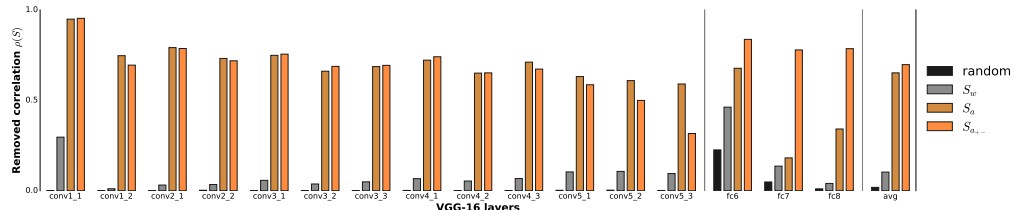

Figure 3: Evaluating $\rho(S)$ for VGG-16 on ImageNet. Higher values are better. The gradient ($S_{\boldsymbol{w}}$), linear estimator ($S_{\boldsymbol{a}}$) and nonlinear estimator ($S_{\boldsymbol{a}_{+-}}$) are compared. An estimator using random directions is the baseline. The network has 5 blocks with 2/3 convolutional layers and 1 max-pooling layer each, followed by 3 dense layers.

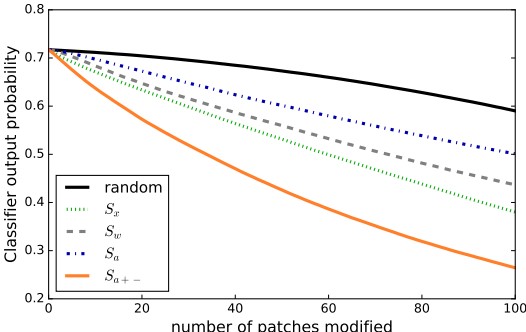

Figure 4: Image degradation experiment on all 50.000 images in the ImageNet validation set. The effect on the classifier output is measured. A steeper decrease is better.

## 5 EXPERIMENTS AND DISCUSSION

To evaluate the quality of the explanations, we focus on the task of image classification. Nevertheless, our method is not restricted to networks operating on image inputs. We used Theano (Bergstra et al., 2010) and Lasagne (Dieleman et al., 2015) for our implementation. We restrict the analysis to the well-known ImageNet dataset (Russakovsky et al., 2015) using the pre-trained VGG-16 model (Simonyan & Zisserman, 2015). Images were rescaled and cropped to 224x224 pixels. The signal estimators are trained on the first half of the training dataset.

The vector $\boldsymbol{v}$, used to measure the quality of the signal estimator $\rho(\boldsymbol{x})$ in Eq. (1), is optimized on the second half of the training dataset. This enables us to test the signal estimators for generalization. All the results presented here were obtained using the official validation set of 50000 samples. The validation set was not used for training the signal estimators, nor for training the vector $\boldsymbol{v}$ to measure the quality. Consequently our results are obtained on previously unseen data.

The linear and the two component signal estimators are obtained by solving their respective closed form solutions (Eq. (4) and Eq. (8)). With a highly parallelized implementation using 4 GPUs this could be done in 3-4 hours. This can be considered reasonable given that several days are required to train the actual network. The quality of a signal estimator is assessed with Eq. (1). Solving it with the closed form solution is computationally prohibitive since it must be repeated for every single weight vector in the network. Therefore we optimize the equivalent least-squares problem using stochastic mini-batch gradient descent with ADAM Kingma & Ba (2015) until convergence. This was implemented on a NVIDIA Tesla K40 and took about 24 hours per optimized signal estimator.

After learning to explain, individual explanations are computationally cheap since they can be implemented as a back-propagation pass with a modified weight vector. As a result, our method produces explanations at least as fast as the work by Dabkowski & Gal (2017) on real time saliency. However, our method has the advantage that it is not only applicable to image models but is a generalization of the theory commonly used in neuroimaging Haufe et al. (2014).

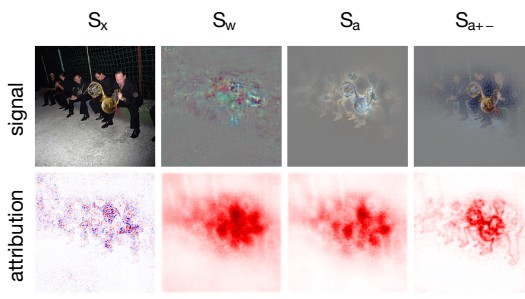

Figure 5: Top: signal. Bottom: attribution. For the trivial estimator $S_x$ the original input is the signal. This is not informative w.r.t. how the network operates.

**Measuring the quality of signal estimators**   In  Fig. 3 we present the results from the correlation measure $\rho(\boldsymbol{x})$, where higher values are better. We use random directions as baseline signal estimators. Clearly, this approach removes almost no correlation. The filter-based estimator $S_{\boldsymbol{w}}$ succeeds in removing some of the information in the first layer. This indicates that the filters are similar to the patterns in this layer. However, the gradient removes much less information in the higher layers. Overall, it does not perform much better than the random estimator. *This implies that the weights do not correspond to the detected stimulus in a neural network.* Hence the implicit assumptions about the signal made by DeConvNet and Guided BackProp is not valid. The optimized estimators remove much more of the correlations across the board. For convolutional layers, $S_{\boldsymbol{a}}$ and $S_{\boldsymbol{a}+-}$ perform comparably in all but one layer. The two component estimator $S_{\boldsymbol{a}+-}$ is best in the dense layers.

**Image degradation**   The first experiment was a direct measurement of the quality of the signal estimators of individual neurons. The second one is an indirect measurement of the quality, but it considers the whole network. We measure how the prediction (after the soft-max) for the initially selected class changes as a function of corrupting more and more patches based on the ordering assigned by the attribution (see Samek et al., 2016). This is also related to the work by Zintgraf et al. (2017). In this experiment, we split the image in non-overlapping patches of 9x9 pixels. We compute the attribution and sum all the values within a patch. We sort the patches in decreasing order based on the aggregate heat map value. In step $n = 1..100$ we replace the first $n$ patches with the their mean per color channel to remove the information in this patch. Then, we measure how this influences the classifiers output. We use the estimators from the previous experiment to obtain the function-signal attribution heat maps for evaluation. A steeper decay indicates a better heat map.

Results are shown in  Fig. 4. The baseline, in which the patches are randomly ordered, performs worst. The linear optimized estimator $S_{\boldsymbol{a}}$ performs quite poorly, followed by the filter-based estimator $S_{\boldsymbol{w}}$. The trivial signal estimator $S_{\boldsymbol{x}}$ performs just slightly better. However, the two component model $S_{\boldsymbol{a}+-}$ leads to the fastest decrease in confidence in the original prediction by a large margin. Its excellent quantitative performance is also backed up by the visualizations discussed next.

**Qualitative evaluation**   In  Fig. 5, we compare all signal estimators on a single input image. For the trivial estimator $S_{\boldsymbol{x}}$, the signal is by definition the original input image and, thus, includes the distractor. Therefore, its noisy attribution heat map shows contributions that cancel each other in the neural network. The $S_{\boldsymbol{w}}$ estimator captures some of the structure. The optimized estimator $S_{\boldsymbol{a}}$ results in slightly more structure but struggles on color information and produces dense heat maps. The two component model $S_{\boldsymbol{a}+-}$ on the right captures the original input during signal estimation and produces a crisp heat map of the attribution.

Fig. 6 shows the visualizations for six randomly selected images from ImageNet. PatternNet is able to recover a signal close to the original without having to resort to the inclusion of additional rectifiers in contrast to DeConvNet and Guided BackProp. We argue that this is due to the fact that the optimization of the pattern allows for capturing the important directions in input space. This contrasts with the commonly used methods DeConvNet, Guided BackProp, LRP and DTD, for which the correlation experiment indicates that their implicit signal estimator cannot capture

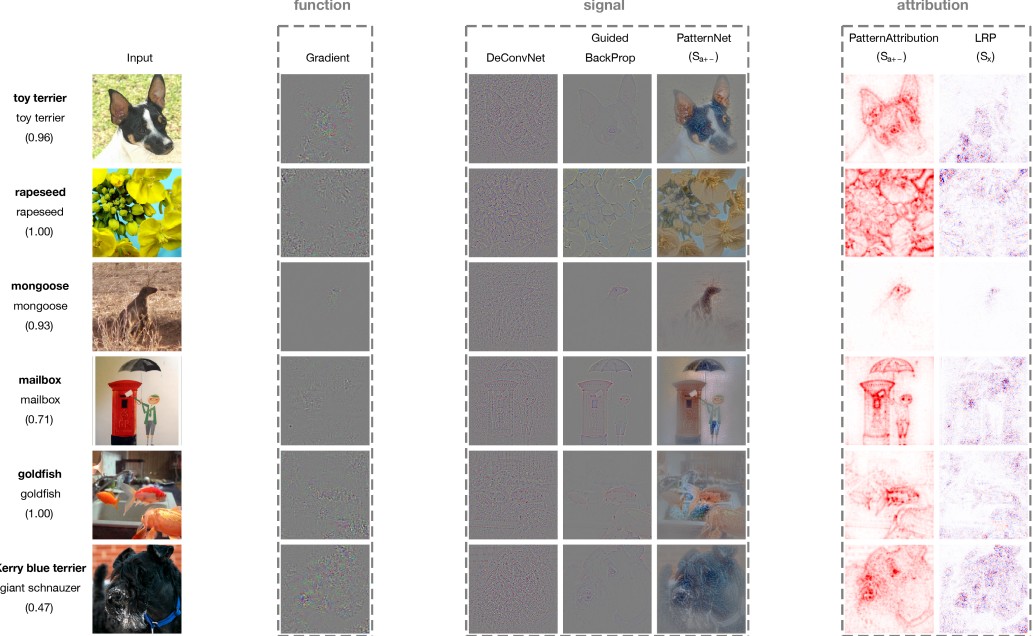

Figure 6: Visualization of random images from ImageNet (validation set). In the leftmost shows column the ground truth, the predicted label and the classifier's confidence. Methods should only be compared within their group. PatternNet, Guided Backprop, DeConvNet and the Gradient (saliency map) are back-projections to input space with the original color channels. They are normalized using $x_{norm} = \frac{x}{2\max|x|} + \frac{1}{2}$ to maximize contrast. LRP and PatternAttribution are heat maps showing pixel-wise contributions.

the true signal in the data. Overall, the proposed approach produces the most crisp visualization in addition to being measurably better, as shown in the previous section. Additonally, we also contrast our methods to the prediction-differences analysis by Zintgraf et al. (2017) in the supplementary material.

**Relation to previous methods** Our method can be thought of as a generalization of the work by Haufe et al. (2014), making it applicable on deep neural networks. Remarkably, our proposed approach can solve the toy example in section 2 optimally while none of the previously published methods for deep learning are able to solve this (Bach et al., 2015; Montavon et al., 2017; Smilkov et al., 2017; Sundararajan et al., 2017; Zintgraf et al., 2017; Dabkowski & Gal, 2017; Zeiler & Fergus, 2014; Springenberg et al., 2015). Our method shares the idea that to explain a model properly one has to learn how to explain it with Zintgraf et al. (2017) and Dabkowski & Gal (2017). Furthermore, since our approach is after training just as expensive as a single back-propagation step, it can be applied in a real-time context, which is also possible for the work done by Dabkowski & Gal (2017) but not for Zintgraf et al. (2017).

## 6 CONCLUSION

Understanding and explaining nonlinear methods is an important challenge in machine learning. Algorithms for visualizing nonlinear models have emerged but theoretical contributions are scarce. We have shown that the direction of the model gradient does not necessarily provide an estimate for the signal in the data. Instead it reflects the relation between the signal direction and the distracting noise contributions ( Fig. 2). This implies that popular explanation approaches for neural networks (DeConvNet, Guided BackProp, LRP) do not provide the correct explanation, even for a simple linear model. Our reasoning can be extended to nonlinear models. We have proposed an objective function for neuron-wise explanations. This can be optimized to correct the signal visualizations (PatternNet) and the decomposition methods (PatternAttribution) by taking the data distribution into

account. We have demonstrated that our methods constitute a theoretical, qualitative and quantitative improvement towards understanding deep neural networks.

ACKNOWLEDGMENTS

This project has received funding from the European Union's Horizon 2020 research and innovation programme under the Marie Sklodowska-Curie grant agreement NO 657679, the BMBF for the Berlin Big Data Center BBDC (01IS14013A), a hardware donation from NVIDIA. We thank Sander Dieleman, Jonas Degraeve, Ira Korshunova, Stefan Chmiela, Malte Esders, Sarah Hooker, Vincent Vanhoucke for their comments to improve this manuscript. We are grateful to Chris Olah and Gregoire Montavon for the valuable discussions.

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

## A  ALGORITHMS

In this section we will give an overview of the visualization algorithms to clarify their actual implementation for ReLu networks. This shows the similarities and the differences between all approaches. For all visualization approaches, the back-projection through a max-pooling layer is only through the path that was active in the forward pass.

## A.1 FUNCTION VISUALISATION

### A.1.1 GRADIENT WITH RESPECT TO THE INPUT

**Initialization** To compute the gradient of an output neuron w.r.t. to the input, we can initialize the gradient at the output neuron with a single 1, for the non-selected output neurons we initialize it at 0.

$$g_i^{output} = y, \qquad g_{j \neq i}^{output} = 0.$$

**Linear or convolutional layer** If it is a linear layer, the gradients from neuron $i$ in layer $l$ can be projected back to its input along the weight vector

$$\boldsymbol{g}^{l-1,i} = \boldsymbol{w} g_i^l.$$

The gradient of neuron $i$ at layer $l-1$ is the sum of all incoming gradients from the layer above:

$$g_i^{l-1} = \sum_j g_i^{l-1,j}.$$

**ReLu layer** To propagate through a ReLu, the propagation is blocked if the ReLu was not active in the forward pass.

$$g_i^{l-1} = \begin{cases} g_i^l & x_i^l > 0 \\ 0 & \text{otherwise} \end{cases}$$

## A.2 SIGNAL VISUALIZATION

### A.2.1 DECONVNET

The DeConvNet visualization is highly similar to the computation of the gradient. The key difference is how the ReLu's are treated.

**Initialization** To compute the visualization of the signal of an output neuron w.r.t. to the input, we can initialize it at the output neuron with the output value $y$, for the non-selected output neurons we initialize it at 0.

$$s_i^{output} = y, \qquad s_{j \neq i}^{output} = 0.$$

**Linear or convolutional layer** If it is a linear layer, the signal from neuron $i$ in layer $l$ can be projected back to its input along the weight vector.

$$\boldsymbol{s}^{l-1,i} = \boldsymbol{w} s_i^l.$$

The signal of neuron $i$ at layer $l-1$ is the sum of all incoming signals from the layer above:

$$s_i^{l-1} = \sum_j s_i^{l-1,j}.$$

**ReLu layer** To propagate through a ReLu, the propagation is blocked if the signal is negative

$$s_i^{l-1} = \begin{cases} s_i^l & s_i^l > 0 \\ 0 & \text{otherwise} \end{cases}$$

### A.2.2 GUIDED BACKPROPAGATION

The Guided Backpropagation visualization is highly similar to the computation of the gradient. and to DeConvNet. The only difference is how the ReLu's are treated.

**Initialization** To compute the visualization of the signal of an output neuron w.r.t. to the input, we can initialize it at the output neuron with the output value $y$, for the non-selected output neurons we initialize it at 0.

$$s_i^{output} = y, \qquad s_{j \neq i}^{output} = 0.$$

**Linear or convolutional layer** If it is a linear layer, the signal from neuron $i$ in layer $l$ can be projected back to its input along the weight vector

$$\boldsymbol{s}^{l-1,i} = \boldsymbol{w} s_i^l.$$

The signal of neuron $i$ at layer $l-1$ is the sum of all incoming signals from the layer above:

$$s_i^{l-1} = \sum_j s_i^{l-1,j}.$$

**ReLu layer** To propagate through a ReLu, the propagation is blocked if the signal is negative or if the neuron was not active in the forward pass

$$s_i^{l-1} = \begin{cases} s_i^l & s_i^l > 0 \ \text{and} \ x_i^l > 0 \\ 0 & \text{otherwise} \end{cases}$$

### A.2.3 PATTERNNET

The PatternNet computation is analogous to the gradient computation. The key difference is that during the backward pass patterns are used instead of weights.

**Initialization** To compute the visualization of the signal of an output neuron w.r.t. to the input, we can initialize it at the output neuron with the output value $y$, for the non-selected output neurons we initialize it at 0.

$$s_i^{output} = y, \qquad s_{j \neq i}^{output} = 0.$$

**Linear or convolutional layer** If it is a linear layer, the signal from neuron $i$ in layer $l$ can be projected back to its input along the pattern vector.

$$\boldsymbol{s}^{l-1,i} = \boldsymbol{a} s_i^l.$$

In our case where we use the $S_{\boldsymbol{a}+-}(\boldsymbol{x})$ estimator, this vector is always the positive components $\boldsymbol{a}_+$ since non-active ReLu's block the propagation.

The signal of neuron $i$ at layer $l-1$ is the sum of all incoming signals from the layer above:

$$s_i^{l-1} = \sum_j s_i^{l-1,j}.$$

**ReLu layer** To propagate through a ReLu, the propagation is blocked if the neuron was not active in the forward pass

$$s_i^{l-1} = \begin{cases} s_i^l & x_i^l > 0 \\ 0 & \text{otherwise} \end{cases}$$

## A.3 ATTRIBUTION VISUALIZATION

### A.3.1 DEEP-TAYLOR DECOMPOSITION

**Initialization** To compute the attribution of an output neuron, we can initialize it at the output neuron with the output value $y$, for the non-selected output neurons we initialize it at 0.

$$r_i^{output} = y, \qquad r_{j \neq i}^{output} = 0.$$

**Linear or convolutional layer** If it is a linear layer, the signal from neuron $i$ in layer $l$ can be projected back to its input w.r.t. a reference point $\boldsymbol{x}_0$

$$\boldsymbol{r}^{l-1,i} = \frac{\boldsymbol{w} \odot (\boldsymbol{x} - \boldsymbol{x}_0)}{\boldsymbol{w}^T \boldsymbol{x}} r_i^l.$$

In this work we use the $S_{\boldsymbol{a}+-}(\boldsymbol{x})$ estimator. Since a non-active ReLu blocks the propagation, we only have to use the positive component $\boldsymbol{a}_+$.

The attribution of neuron $i$ at layer $l-1$ is the sum of all incoming attributions from the layer above:

$$r_i^{l-1} = \sum_j r_i^{l-1,j}.$$

**ReLu layer**  To propagate through a ReLu, the propagation is blocked if the ReLu was not active in the forward pass.

$$r_i^{l-1} = \begin{cases} r_i^l & x_i^l > 0 \\ 0 & \text{otherwise} \end{cases}$$

### A.3.2  LRP

LRP corresponds to $\boldsymbol{x}_0 = \boldsymbol{0}$. This allows us to re-write the distribution rule for the linear and convolutional layers as:

$$\boldsymbol{r}^{l-1,i} = \frac{\boldsymbol{w} \odot \boldsymbol{x}}{\boldsymbol{w}^T \boldsymbol{x}} r_i^l.$$

Division by zero is not an issue since the propagation is blocked at the ReLu if $\boldsymbol{w}^T \boldsymbol{x} <= 0$.

### A.3.3  PATTERNATTRIBUTION WITH $S_{\boldsymbol{a}+-}(\boldsymbol{x})$

PatternAttributions corresponds to $\boldsymbol{x}_0 = \boldsymbol{x} - S_{\boldsymbol{a}+-}(\boldsymbol{x})$. Because propagation is blocked at the ReLu if $\boldsymbol{w}^T \boldsymbol{x} <= 0$, we can always use $\boldsymbol{x}_0 = \boldsymbol{x} - \boldsymbol{a}_+ \boldsymbol{w}^T \boldsymbol{x}$.

This allows us to re-write the distribution rule for the linear and convolutional layers as:

$$\boldsymbol{r}^{l-1,i} = \frac{\boldsymbol{w} \odot \left(\boldsymbol{x} - \boldsymbol{x} + \boldsymbol{a}_+ \boldsymbol{w}^T \boldsymbol{x}\right)}{\boldsymbol{w}^T \boldsymbol{x}} r_i^l.$$

This can be simplified to

$$\boldsymbol{r}^{l-1,i} = \boldsymbol{w} \odot \boldsymbol{a}_+ r_i^l$$

since division by zero is not an issue because the propagation is blocked at the ReLu if $\boldsymbol{w}^T \boldsymbol{x} <= 0$.

# B   QUALITATIVE COMPARISON TO PREDICTION-DIFFERENCES ANALYSIS

To create the Predictive-Differences analysis visualizations Zintgraf et al. (2017), we used the open-source code provided by the authors with the default parameter settings provided for VGG.

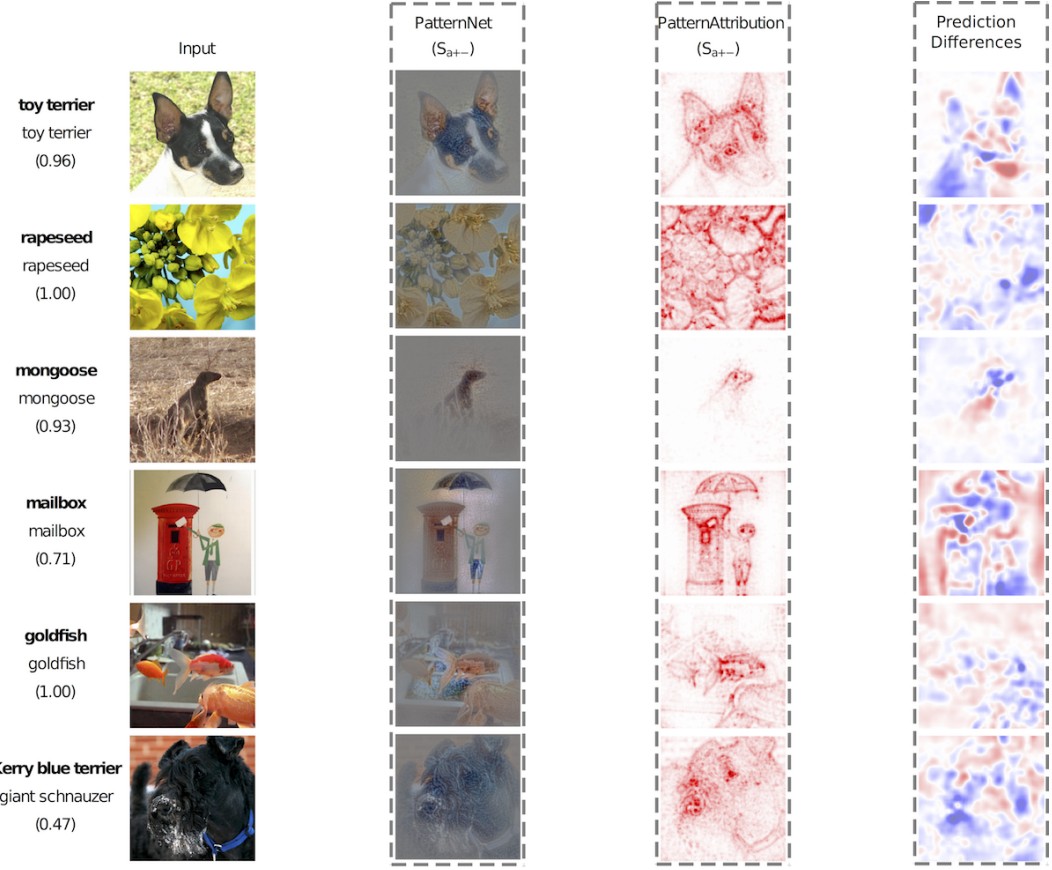

Figure 7: Visualization of random images from ImageNet (validation set). In the leftmost shows column the ground truth, the predicted label and the classifier's confidence. Comparison between the proposed methods PatternNet and PatternAttribution to the Prediction-Differences approach by Zintgraf et al. (2017).

