# OpenReview forum: "Learning how to explain neural networks: PatternNet and PatternAttribution"
_ICLR.cc/2018/Conference — Accept (Poster)_

### Official Review · AnonReviewer3 · 2017-11-27
**Very interesting and clear work**

**Rating:** 8
**Confidence:** 4

**Review:**

The authors analyze show theoretical shortcomings in previous methods of explaining neural networks and propose an elegant way to remove these shortcomings in their methods PatternNet and PatternAttribution.

The quest of visualizing neural network decision is now a very active field with many contributions. The contribution made by the authors stands out due to its elegant combination of theoretical insights and improved performance in application. The work is very detailed and reads very well.

I am missing at least one figure with comparison with more state-of-the-art methods (e.g. I would love to see results from the method by Zintgraf et al. 2017 which unlike all included prior methods seems to produce much crisper visualizations and also is very related because it learns from the data, too).

Minor questions and comments:
* Fig 3: Why is the random method so good at removing correlation from fc6? And the S_w even better? Something seems special about fc6.
* Fig 4: Why is the identical estimator better than the weights estimator and that one better than S_a?
* It would be nice to compare the image degradation experiment with using the ranking provided by the work from Zintgraf which should by definition function as a kind of gold standard
* Figure 5, 4th row (mailbox): It looks like the umbrella significantly contributes to the network decision to classify the image as "mailbox" which doesn't make too much sense. Is is a problem of the visualization  (maybe there is next to no weight on the umbrella), of PatternAttribution or a strange but interesting a artifact of the analyzed network?
* page 8 "... closed form solutions (Eq (4) and Eq. (7))" The first reference seems to be wrong. I guess Eq 4. should instead reference the unnumbered equation after Eq. 3.

Update 2018-01-12: Upgraded Rating from 7 to 8 (see comment below)

---

> ### Author Response · Authors · 2018-01-05
> **Response**
>
> We thank the reviewer for his detailed comments!
>
> Quote:
> I am missing at least one figure with comparison with more state-of-the-art methods (e.g. I would love to see results from the method by Zintgraf et al. 2017 which unlike all included prior methods seems to produce much crisper visualizations and also is very related because it learns from the data, too).
>
> Answer
> The reason the Zintgraf paper was left out initially is that it does not perform an exact decomposition of the output value into input contributions  such as defined by LRP and DTD. Instead it defines a different, bayesian, measure on importance. The visualisations are therefore not directly comparable.  The work by Zintgraf is excellent and we have included a comparison for the qualitative visualisation in the appendix.
>
>
>
>
> Quote:
> * Fig 3: Why is the random method so good at removing correlation from fc6? And the S_w even better? Something seems special about fc6.
>
> Answer:
> The weight vector always has a dot product of 1 with the informative direction. They do not coincide but they are correlated. Therefore it is to be expected that S_w performs better than the random direction. The fc_6 result is indeed surprising. It is the first fully connected layer and has therefore the largest dimensionality in the input. This makes measuring the quality of the signal estimators more difficult in this layer.
>
>
>
>
> Quote:
> * Fig 4: Why is the identical estimator better than the weights estimator and that one better than S_a?
>
> Answer:
> The degradation experiment favors the identity estimator since LRP(=identity estimator) reduces to gradient x input. If the gradient would be constant (as it is in a linear model) the biggest change you can create in the logit by changing a single dimension.That the linear pattern does not work as well as the two-component version can be attributed to the fact that it incorrectly models the signal component by ignoring nonlinear component introduced by the ReLu.
>
>
>
>
> Quote:
> * It would be nice to compare the image degradation experiment with using the ranking provided by the work from Zintgraf which should by definition function as a kind of gold standard
>
> Answer:
> It would be interesting but not feasible unfortunately. The work by Zintgraf et al takes for VGG 70 minutes per image (according to their paper). Processing each of the 50.000 validation images already takes 50.000 images  * 70 min /60 (min) /24 (hours) = 2430 days of compute. Since we have to process each image multiple times times this is simply not possible.
>
>
>
>
> Quote:
> * Figure 5, 4th row (mailbox): It looks like the umbrella significantly contributes to the network decision to classify the image as "mailbox" which doesn't make too much sense. Is is a problem of the visualization  (maybe there is next to no weight on the umbrella), of PatternAttribution or a strange but interesting a artifact of the analyzed network?
>
> Answer:
> We have no definitive explanation yet. A possible explanation is that the second to last layer has to contain information on all the classes. Umbrella is one of these classes. Also in the explanation of the Zintgraf method the umbrella contributes positively to the mailbox class. This is an indication that it could be an artifact of the analyzed network.
>
>
>
>
> Quote:
> * page 8 "... closed form solutions (Eq (4) and Eq. (7))" The first reference seems to be wrong. I guess Eq 4. should instead reference the unnumbered equation after Eq. 3.
>
> Answer:
> Thanks: we will update the manuscript.

---

> > ### Comment · AnonReviewer3 · 2018-01-12
> > **Response to Response**
> >
> > Thank you for your detailed response.
> >
> > I appreciate the added comparison with the Zintgraf method in the appendix. To me, the attribution often looks surprisingly different (ignoring the obvious differnence that only the Zintgraf method can output negative attribution).
> >
> > Regarding the image degradation experiment in Figure 4: I understand that it is not feasible to run the Zintgraf method on the full validation dataset, but I still
> >  think it would be interesting to use it as a gold standard at least on a small subset of images. Since the paper is now already one year old, chances are that the
> >  computation times are shorter on modern hardware. But arguably it is still not feasible on a reasonbly representative dataset, which is very unfortunate.
> >
> > The newly added summary of the different methods in the appendix is very helpful, thankyou for that!
> >
> > By the way, do you plan to release the code? I would be interested in applying your method to a few of my networks.
> >
> > To summarize, I am very happy with the paper now and will upgrade my rating to "8: Top 50% of papers".

---

> > > ### Public Comment · (anonymous) · 2018-01-20
> > > **Other missing baselines**
> > >
> > > Both integrated gradients [1] and DeepLift [2] are good, commonly used methods which are certainly fast enough to be used in the image degradation experiment. A meaningful comparison to them, or any method published within the past two years, is very clearly missing from this paper.
> > >
> > > Context: I'm an active researcher in the space, was going to post an unsolicited, highly negative review, due to the poor comparison with state of the art, but couldn't find the time.
> > >
> > > [1] https://arxiv.org/abs/1703.01365
> > > [2] https://arxiv.org/abs/1704.02685

---

### Official Review · AnonReviewer1 · 2017-11-27
**Important framework, tools, and criterion for understanding deep neural networks**

**Rating:** 8
**Confidence:** 3

**Review:**

summary of article:
This paper organizes existing methods for understanding and explaining deep neural networks into three categories based on what they reveal about a network: functions, signals, or attribution. “The function extracts the signal from the data by removing the distractor. The attribution of output values to input dimensions shows how much an individual component of the signal contributes to the output…” (p. 5). The authors propose a novel quality criterion for signal estimators, inspired by the analysis of linear models. They also propose two new explanatory methods, PatternNet (for signal estimation) and PatternAttribution (for relevance attribution), based on optimizing their new quality criterion. They present quantitative and qualitative analyses comparing PatternNet and PatternAttribution to several existing explanation methods on VGG-19.

* Quality: The claims of the paper are well supported by quantitative results and qualitative visualizations.
* Clarity: Overall the paper is clear and well organized. There are a few points that could benefit from clarification.
* Originality: The paper puts forth an original framing of the problem of explaining deep neural networks. Related work is appropriately cited and compared. The authors's quality criterion for signal estimators allows them to do a quantitative analysis for a problem that is often hard to quantify.
* Significance: This paper justifies PatternNet and PatternAttribution as good methods to explain predictions made by neural networks. These methods may now serve as an important tool for future work which may lead to new insights about how neural networks work.

Pros:
* Helps to organize existing methods for understanding neural networks in terms of the types of descriptions they provide: functions, signals or attribution.
* Creative quantitative analyses that evaluate their signal estimator at the level of single units and entire networks.

Cons:
* Experiments consider only the pre-trained VGG-19 model trained on ImageNet. Results may not generalize to other architectures/datasets.
* Limited visualizations are provided.

Comments:
* Most of the paper is dedicated to explaining these signal estimators and quality criterion in case of a linear model. Only one paragraph is given to explain how they are used to estimate the signal at each layer in VGG-19. On first reading, there are some ambiguities about how the estimators scale up to deep networks. It would help to clarify if you included the expression for the two-component estimator and maybe your quality criterion for an arbitrary hidden unit.
* The concept of signal is somewhat unclear. Is the signal
    * (a) the part of the input image that led to a particular classification, as described in the introduction and suggested by the visualizations, in which case there is one signal per image for a given trained network?
    *  (b) the part of the input that led to activation of a particular unit, as your unit wise signal estimators are applied, in which case there is one signal for every unit of a trained network? You might benefit from two terms to separate the unit-level signal (what caused the activation of a particular unit?) from the total signal (what caused all activations in this network?).
* Assuming definition (b) I think the visualizations would be more convincing if you showed the signal for several output units. One would like to see that the signal estimation is doing more than separating foreground from background but is actually semantically specific. For instance, for the mailbox image, what does the signal look like if you propagate back from only the output unit for umbrella compared to the output unit for mailbox?
* Do you have any intuition about why your two-component estimator doesn’t seem to be working as well in the convolutional layers? Do you think it is related to the fact that you are averaging within feature maps? Is it strictly necessary to do this averaging? Can you imagine a signal estimator more specifically designed for convolutional layers?

Minor issues:
* The label "Figure 4" is missing. Only subcaptions (a) and (b) are present.
* Color scheme of figures: Why two oranges? It’s hard to see the difference.

---

> ### Author Response · Authors · 2018-01-05
> **Response**
>
> We thank the reviewer for the in depth and careful review!
>
>
> Quote:
> * Most of the paper is dedicated to explaining these signal estimators and quality criterion in case of a linear model. Only one paragraph is given to explain how they are used to estimate the signal at each layer in VGG-19. On first reading, there are some ambiguities about how the estimators scale up to deep networks. It would help to clarify if you included the expression for the two-component estimator and maybe your quality criterion for an arbitrary hidden unit.
>
> Answer:
> The two component estimator is in Eq. 4 with the direction defined as by Eq. 7. The Quality criterion is eq. 1. Since we analyze neurons between non-linearities. In the manuscript we focussed on ReLu networks, but ideally different estimators will be developed for different non-linearities in the future.
> The algorithm for the back-propagation will be added to the manuscript in the appendices.
>
>
>
>
> Quote:
> * The concept of signal is somewhat unclear. Is the signal
>     * (a) the part of the input image that led to a particular classification, as described in the introduction and suggested by the visualizations, in which case there is one signal per image for a given trained network?
>     *  (b) the part of the input that led to activation of a particular unit, as your unit wise signal estimators are applied, in which case there is one signal for every unit of a trained network? You might benefit from two terms to separate the unit-level signal (what caused the activation of a particular unit?) from the total signal (what caused all activations in this network?).
>
> Answer:
> In our analysis we used definition b and define it neuron-wise. As mentioned in the manuscript, the visualised signal is a superposition of what are assumed to be the neuron-wise signals.
>
>
>
>
> Quote:
> * Do you have any intuition about why your two-component estimator doesn’t seem to be working as well in the convolutional layers? Do you think it is related to the fact that you are averaging within feature maps? Is it strictly
> We have no definitive explanation and are still investigating this.

---

### Official Review · AnonReviewer2 · 2017-11-27
**Interesting, but premature contribution on interpretability**

**Rating:** 6
**Confidence:** 4

**Review:**

I found this paper an interesting read for two reasons: First, interpretability is an increasingly important problem as machine learning models grow more and more complicated. Second, the paper aims at generalization of previous work on confounded linear model interpretation in neuroimaging (the so-called filter versus patterns problem). The problem is relevant for discriminative problems: If the objective is really to visualize the generative process,  the "filters" learned by the discriminative process need to be transformed to correct for spatial correlated noise.

Given the focus on extracting visualization of the generative process, it would have been meaningful to place the discussion in a greater frame of generative model deep learning (VAEs, GANs etc etc). At present the "state of the art" discussion appears quite narrow, being confined to recent methods for visualization of discriminative deep models.

The authors convincingly demonstrate for the linear case, that their "PatternNet" mechanism can produce the generative process (i.e. discard spatially correlated "distractors"). The PatternNet is generalized to multi-layer ReLu networks by construction of node-specific pattern vectors and back-propagating these through the network. The "proof" (eqs. 4-6) is sketchy and involves uncontrolled approximations. The back-propagation mechanism is very briefly introduced and depicted in figure 1.

Yet, the results are rather convincing. Both the anecdotal/qualitative examples and the more quantitative patch elimination experiment figure 4a (?number missing)

I do not understand the remark: "However, our method has the advantage that it is not only applicable to image models but is a generalization of the theory commonly used in neuroimaging Haufe et al. (2014)."  what ??

Overall, I appreciate the general idea. However, the contribution could have been much stronger based on a detailed derivation with testable assumptions/approximations, and if based on a clear declaration of the aim.

---

> ### Author Response · Authors · 2018-01-05
> **Response**
>
> We thank the reviewer for the insightful and balanced review.
>
>
>
>
> Quote:
> Given the focus on extracting visualization of the generative process, it would have been meaningful to place the discussion in a greater frame of generative model deep learning (VAEs, GANs etc etc). At present the "state of the art" discussion appears quite narrow, being confined to recent methods for visualization of discriminative deep models.
>
> Answer:
> We intentionally kept the scope of the state of the art focussed on the methods for discriminative models. We motivate this choice as follows: our analysis focuses on these methods that analyse discriminative models. The goal of the interpretability methods is to find what the informative component is in the data. The general field of generative modelling on the other hand tries to model the full data, both the informative and non-informative components.
> We refrained from directly comparing to the greater framework of generative models in deep learning because we wanted to prevent confusion among the readers and to limit the length of the manuscript. That being said, we do believe that these models (GAN’s, VAE, …) can become part of methods for interpretability, e.g. they could be used for signal estimation.
>
>
>
>
> Quote:
> The authors convincingly demonstrate for the linear case, that their "PatternNet" mechanism can produce the generative process (i.e. discard spatially correlated "distractors"). The PatternNet is generalized to multi-layer ReLu networks by construction of node-specific pattern vectors and back-propagating these through the network. The "proof" (eqs. 4-6) is sketchy and involves uncontrolled approximations. The back-propagation mechanism is very briefly introduced and depicted in figure 1.
>
> Answer:
> To obtain PatternNet, we started by maximizing equation 1.
> This equation describes that we want to remove the signal from the input.
> The signal being the component in the input that is predictive (linearly) about the output of the neuron.
> In Equation 2, we show that this can be done by ensuring that the covariance between the signal and the output is identical to the covariance between the original input and the output. This is shown generally without making additional assumptions on the signal.
>
> However, to turn Eq. 2 into an actionable approach, we must make an assumption on the functional form of the signal estimator.  This is what we do in Eq. 3 for the linear case and Eq. 4 for the non-linear case.
> On the other hand, Equations 5 and 6 are simply re-writing the covariance. There is no additional approximation.
> The step to Eq. 7 introduces a new assumption. Here we assume that the contribution to the covariances for 5 and 6 are equal in the non-firing regime are equal. The same holds for the firing (activation above 0) regime. Since this is an approximation, we carefully designed our experiments (including the one in Fig. 3) to measure the quality of this approximation.
>
> We do agree with the reviewer that it would be extremely valuable to create a formal proof that this is the optimal approach. However, so far we did not find a way to create this proof. Furthermore, we are not aware of any formal approach within the field of interpretability that is able to do this. Instead, we have to rely on the evidence that (1) our approach can solve the linear toy problem correctly and (2) our experimental results indicate that it is a quantitative and qualitative improvement over previous methods.
> To clarify the back-propagation mechanism we updated the manuscript with an appendix making the algorithms explicit.
>
>
>
>
> Quote:
> Yet, the results are rather convincing. Both the anecdotal/qualitative examples and the more quantitative patch elimination experiment figure 4a (?number missing)
>
> Answer:
> We will update the caption of the figure.
>
>
>
>
> Quote:
> I do not understand the remark: "However, our method has the advantage that it is not only applicable to image models but is a generalization of the theory commonly used in neuroimaging Haufe et al. (2014)."  what ??
>
> Answer:
> We will rephrase this as: Our method is a generalization of the analysis of linear models known in Neuroimaging (Haufe et al. (2014)) that makes it applicable to deep networks.

---

### Public Comment · (anonymous) · 2017-10-25
**Doesn't integrated gradients already do this?**

Doesn't integrated gradients, which you cite (https://arxiv.org/abs/1703.01365, ICML 2017), produce the theoretically correct explanation for a linear model, and also prove that their method is the only method which can do so under pretty reasonable assumptions?

---

> ### Public Comment · (anonymous) · 2017-11-03
> **I don't think it does**
>
> In a linear model the gradient is constant, so the integrated gradients method produces the element-wise product of the gradient and the image (assuming an all-zero baseline).
>
> My understanding is that the authors here make the point that the image is generally signal + distractors (see Fig. 2), and the correct attribution would be the element-wise product of the gradient and the signal (without distractors). Thus, you need to first estimate what's the signal, which is what the authors address here.

---

### Public Comment · (anonymous) · 2017-10-30
**Baselines?**

Why don't you baseline against prior work in your evaluation?

There's a big problem with clutter in this space, with lots of other approaches to doing this, even that have been published just this year. It's incumbent upon you to demonstrate that your method is better than the 10 other methods you cite, rather than tucking away comparisons into a one-paragraph section at the end of your results, which reads more like a related work section.

---

### Public Comment · (anonymous) · 2022-07-20
**Signal-distractor decomposition is ill-defined**

Thanks for the interesting and thought-provoking paper!

As the paper already mentions, it is an ill-posed problem to extract signal and distractor components from a linear model, absent further assumptions such as orthogonality or statistical independence among the signal and distractor vectors. Further, the proposed signal quality estimator is also unable to identify a unique decomposition.

For example, consider a linear model on a 2D dataset. One can always write an orthogonal basis vector decomposition, where the signal is  orthogonal to the weight vector and the distractor is along the weight vector. In the 2D example, given a weight vector w = [1, -1], we can project any data point x along distractor d = [1, 1] and signal s = [1, - 1], and this decomposition identifies the correct distractor and hence produces high scores under the proposed signal quality estimator of equation 1.

Given this non-uniqueness, it is incorrect to state that linear models do not recover the "theoretically correct" decomposition. As shown above, linear models also provide an estimate of signal / distractor that is compatible with equation 1, and are thus not "incorrect" in any way.

---

### Decision · Program_Chairs · 2018-01-29
**ICLR 2018 Conference Acceptance Decision**

**Decision:**

Accept (Poster)

**Comment:**

The paper shows that many of the current state-of-the-art interpretability methods are inaccurate even for linear models. Then based on their analysis of linear models they propose a technique that is thus accurate for them and also empirically provides good performance for non-linear models such as DNNs.